# Empirical Analysis of the Status and Influencing Factors of Catastrophic Health Expenditure of Migrant Workers in Western China

**DOI:** 10.3390/ijerph16050738

**Published:** 2019-03-01

**Authors:** Li Liu, Xuewen Zhang, Longchao Zhao, Ningxiu Li

**Affiliations:** 1West China School of Public Health, Sichuan University, Chengdu 610041, China; liulivita@163.com (L.L.); longchaozhao@163.com (L.Z.); 2School of Public Health, Jining Medical University, Jining 272067, China; zhangxuewen324@163.com

**Keywords:** catastrophic health expenditure, migrant workers, influencing factors, logistic regression, multiple correspondence analysis

## Abstract

*Objective:* To understand the current situation and influencing factors of catastrophic health expenditure (CHE) of migrant workers in Western China. *Method:* Sample data were obtained by cluster random sampling. Data were entered and sorted using Epidata 3.1 and SPSS 21.0. The statistical analysis involved a descriptive analysis, chi-square tests, multivariate unconditional logistic regression, and multiple correspondence analysis (MCA). *Results:* A total of 1271 households were surveyed, and the incidence of CHE was 12.5% (159/1271). The multivariate logistic regression showed that households with elderly people over 65 years old (0R = 2.05, 95% CI = 1.42–2.97), children under five years old (0R = 2.61, 95% CI = 1.53–4.48), at least one person with no basic medical insurance (0R = 2.96, 95% CI = 2.08–4.23), chronically ill patients (0R = 1.85, 95% CI = 1.23–2.77), and hospitalized patients (0R = 3.61, 95% CI = 2.31–5.62) contributed to the risk of CHE. Compared to migrant workers in the >30,000 Yuan household per capita annual income group, the 10,001–20,000 Yuan income group (0R = 2.35, 95% CI = 1.44–3.82) and ≤10,000 Yuan income group (0R = 3.72, 95% CI = 2.09–6.62) had a higher risk of CHE occurrence. Compared to migrant workers in the university and above head-of-household education group, those in the primary level or below education group (0R = 5.90, 95% CI = 3.02–11.5) had a higher risk of CHE occurrence. MCA revealed a strong interrelationship between the following risk factors and CHE: household per capita annual income ≤10,000 Yuan, primary school education level or below for the head of the household, and having at least one person in the household with no basic medical insurance. *Conclusions:* CHE incidence amongst migrant workers in Western China is a serious issue, and policymakers should pay more attention to these migrant workers’ households that are more prone to CHE than others, so as to decrease the incidence of CHE in this group.

## 1. Introduction

Out-of-pocket (OOP) payments are the primary source of healthcare financing in many countries, and these are considered “catastrophic” when they drive households to reduce expenditure on basic necessities [1,2]. Thus, the concept of catastrophic health expenditure (CHE) is introduced, which means that within a certain period of time, the proportion of family health expenditure in household non-food expenditure exceeds a certain defined standard, and exceeds the family’s affordable range and leads to a serious decline in living standards [3,4]. Previous studies suggest that the incidence of CHE is higher in those countries where the share of private out-of-pocket payments in total healthcare expenditure is higher than 20% [5,6]. This share varies across European countries, from 6% in the Netherlands to 32% in Portugal [7]. In China, the medical expenses of households make up 29.27% of total expenditure [8], which is a far higher level than the OECD average of 19.6% (2011) [7]. Since most countries are prone to CHE—which is one of the core evaluation indicators of the economic risk of disease and an important indicator for measuring the fairness of healthcare [2,9]—there have been many studies on CHE. At present, research on CHE mainly focuses on vulnerable families, such as rural families [10,11], urban low-income families [12], empty nest elderly families, and so on [13,14].

However, with the rapid development of China’s industrialization and urbanization processes, a large number of surplus rural laborers have migrated to cities and engaged in various non-agricultural occupations in cities and towns. These household registrations remain in rural areas, and individuals who work in the cities in non-agricultural industries for six months or more are collectively referred to as migrant workers [15]. By the end of 2017, the total number of migrant workers in China reached 287 million, making it the world’s largest floating population. This group of families has made great contributions to China’s modernization drive, and this is a catalyst for social progress, but it also breeds a series of new problems [16]. The existing urban and rural dual household registration system has led to “people in the city, households in the countryside”, and these migrant workers are unable to enjoy the basic public health services of urban residents or the basic public health services that they have in the rural areas; thus, health protection has long been in a weak position. Studies have shown that there is a strong correlation between immigration behavior and health status [17,18,19,20]. Coupled with the low income of a family, it is easy to suffer from CHE as a result of the burden of illness or hospitalization leading to poverty, which affects the sustainability of normal life. Studies have been conducted on the existence of related rights in the city, the difficulty of social integration, and the impact of strong material pressure and mental stress [21,22,23]. However, the ability of the group to bear the burden of medical expenses and the level of CHE have not been reported in China or around the world. Therefore, this study was the first to explore the situation and influencing factors of Chinese migrant workers’ family CHE, in order to fill the gap in CHE research on migrant workers, as well as to provide a reference for promoting the citizenization of migrant workers and promoting healthcare fairness.

## 2. Materials and Methods

### 2.1. Study Sites and Participants

A stratified multistage cluster sampling technique was used. In the first stage, the sizes of the floating populations in Chengdu, Sichuan Province, and Kunming, Yunnan Province (the main gathering areas of migrant workers in Western China) were sampled. In the second stage, based on the social and economic conditions, two central districts and one surrounding district from Chengdu, and one central district and one surrounding district from Kunming, were selected. In the third stage, one community from each district in Chengdu, and two communities from each district in Kunming were sampled. In the fourth stage, based on the characteristics of the community population, each community was allocated into 2–5 residential areas. In the fifth stage, according to the number of people living in the residential area, two buildings were selected from each residential area, and all the households of migrant workers in the selected buildings were interviewed in face-to-face surveys.

“Residential houses” were identified as housing migrants versus local urban residents by asking mainly the following questions: (1) Whether the household registration was in the rural areas; (2) whether they had entered the city to work in non-agricultural labor; and (3) whether they had been working in non-agricultural industries for more than 6 months. If the above questions were answered with “yes”, the household was included in our research, and vice versa. We surveyed 1048 households and 223 households in Kunming and Chengdu, respectively, totaling 1271 households.

### 2.2. Data Collection

A cross-sectional study was conducted by our research team from 2014 to 2016. All households of migrant workers were interviewed face to face using a questionnaire answered by the member most familiar with the family’s situation. The questionnaire included items about family size, head-of-household gender, head-of-household age, head-of-household education, whether there were elderly people aged over 65, whether there were children under 5 years old, whether they had basic medical insurance, whether they had chronic diseases, and whether they contained hospitalized patients, as well as other household characteristics, such as family income, family expenditure, household consumption expenditure, household food consumption expenditure, family medical expenditure, etc. Data were collected by trained Master’s or Doctoral students from the West China School of Public Health, Sichuan University.

### 2.3. Ethics Statement

Recruitment in this study was based entirely on the principle of voluntary and informed consent. The medical ethics committee of the Fourth Hospital of West China (West China School of Public Health, Sichuan University) approved this study.

### 2.4. Indicator Calculation

The literature indicates different forms of calculations and cutoff points for estimating CHE, and there is no consensus regarding the most adequate form to be used in studies on this subject [24]. Consequently, in this study, the most commonly used technique termed the Xu method, was used to calculate the CHE index [2,25]. Households with a healthcare expenditure higher than 40% of their capacity to pay were grouped under the category “households facing catastrophic healthcare expenditure”. The capacity to pay refers to the household’s effective income minus its livelihood costs. The effective income is based on the total household expenditure within one specific time period. In many countries, this income measure has been considered to be a better measure than the income reported in household surveys, which represents purchasing power [26]. Xu et al. [2,25] have described this methodology in detail.

### 2.5. Statistical Methods

Data entry was performed using Epidata 3.1. The data were sorted and analyzed using SPSS 21.0. The categorical variables were expressed by frequency and percentage, and the chi-square test was used for comparisons between groups. This study is the first to use a multi-factor unconditional logistic regression model to analyze the risk factors of CHE. Multiple correspondence analysis (MCA) was also used to explore the degree of correlation between the various factors.

The MCA analyzes the category associations between multiple categorical variables. The association between multiple categorical variables can be expressed in a graph, which is not possible under other statistical methods. This was analyzed using the dimension reduction module of SPSS 21.0 [27]. There are two main applicable conditions for the MCA: The cumulative contribution rate of the first two factors should be greater than 75% when analyzing [28,29], and there should be correlations between row variables and column variables [30,31]. The cumulative contribution rate of the first two common factors was explored by grouping variables, and the results from the logistic regression analysis and our professional knowledge were combined to select and eliminate variables. The applicability test uses the chi-square test to perform a pairwise analysis of the variables to be analyzed, and then it selects statistically significant indicators (test level α = 0.05) to be included in the MCA [27,28,29]. The distance and position of various scatter points in the space in the MCA map reflect the relationship between them. If the point of a certain category of a variable is in the same orientation and distance with a certain category of other variables, it indicates that there is a strong correlation, while the reverse would show that the relationship between the two variables is either weak or unrelated [30,31].

## 3. Results

### 3.1. General Situation

A total of 1271 households were surveyed, with an average family population of 2.76, an average household income of 68,494 Yuan, and an average annual living expenditure of 36,435 Yuan, of which food expenditure was 17,210 Yuan and medical expenditure was 2553 Yuan. The number of CHE households was 159, and the incidence of catastrophic health expenditure was 12.5%.

### 3.2. Univariable Analysis of Different Migrant Workers’ Family Characteristics

There were significant differences in the CHE rates in terms of family size, household per capita annual income, head-of-household level of education, elderly people over 65 years old, children under five years old, the presence of at least one person with no basic medical insurance, chronically ill patients, and hospitalized patients (*p* < 0.001). However, no statistically significant differences were found in terms of the head-of-household gender and the head-of-household age between CHE occurrences and non-CHE occurrences (*p* > 0.05) (Table 1).

### 3.3. Multivariate Analysis of CHE of Migrant Workers’ Families

Statistically significant variables from the univariate analysis were included in the multivariate logistic regression model. The results showed that, amongst migrant workers in Western China, a significantly higher CHE risk was observed in household groups with elderly people over 65 years old (0R = 2.05, 95% CI = 1.42–2.97), children under five years old (0R = 2.61, 95% CI = 1.53–4.48), those with at least one person with no basic medical insurance (0R = 2.96, 95% CI = 2.08–4.23), those with chronically ill patients (0R = 1.85, 95% CI = 1.23–2.77), and those with hospitalized patients (0R = 3.61, 95% CI = 2.31–5.62). Compared to migrant workers in the >30,000 Yuan household per capita annual income group, the 10,001–20,000 income group (0R = 2.35, 95% CI = 1.44–3.82) and the ≤10,000 income group (0R = 3.72, 95% CI = 2.09–6.62) had a higher risk of CHE occurrence. Compared to migrant workers in the university and above head-of-household education group, the primary level or below education group (0R = 5.90, 95% CI = 3.02–11.5) had a higher risk of CHE occurrence (Table 2).

### 3.4. Multiple Correspondence Analysis between Factors

Based on the multivariate logistic regression analysis and our expertise, four variables (CHE, household per capita annual income, head-of-household education, and at least one person with no basic medical insurance) were selected for the MCA. The correlation analysis indicated that all the selected factors were correlated with each other (Table 3).

We observed that the contribution rates of the first and second dimensions were 46.1% and 30.5%, respectively, with a cumulative contribution rate of 76.6%; thus, the MCA fitted well. The closer the distance between the scatter points, the more pronounced the association tendency, and accordingly, the points in Figure 1 could be roughly divided into two clusters. From the two clusters we observed that: (i) There was a close relationship between CHE, household per capita annual income ≤10,000 Yuan, primary school or below education, and at least one person with no basic medical insurance; and (ii) there was a close relationship between no CHE, >20,000 Yuan, junior high school–high school education level, and having basic medical insurance.

## 4. Discussion

Migrant workers are more vulnerable to CHE than urban residents. This study found that the incidence rate of migrant workers in the sample area in 2016 was 12.5%, which was higher than that of ordinary urban residents in China by three percentage points (9.5%) [32,33]. From the perspective of the citizenization of migrant workers, this level of difference means that the economic risk-sharing mechanism for migrant workers’ families is not yet sound, the economic risk of disease is high, and healthcare fairness needs to be further improved.

Consistent with other studies [10,11,33,34], our results showed that CHE is concentrated amongst those who are poorer and less fortunate. International evidence has also shown that people of lower economic status are more likely to suffer from serious illness, and may become impoverished due to the medical costs [35]. One reason that low-income households are more affected may be their relatively low purchasing power. However, when there is a need for medical services in a low-income household, low purchasing power does not stop that household from paying for healthcare. This causes high healthcare expenses relative to purchasing ability, which can lead to low-income families experiencing CHE [36]. These findings indicate the importance—especially for low-income migrant worker households—of living and medical aid from municipal governments and other welfare programs. Such programs have the potential to reduce the risk of CHE for the most vulnerable households.

Households with primary education or below have a significantly higher risk of CHE than those with a university degree or above. This is because the educational level affects the family’s health philosophy, and those with a higher education have a higher degree of emphasis on health. More disease prevention and control measures have been taken to prevent the occurrence of major diseases and reduce the probability of CHE [37].

Consistent with previous studies [38,39], our study indicated that households with elderly family members or children under five years old were at high risk of catastrophic health expenditure. This was not surprising since previous studies have already shown that individuals who are elderly or ≤5 years old are the high-risk group, and have a higher disease incidence compared to other population groups. As a result, they need more healthcare services, which makes them more prone to CHE [40,41].

The primary policy aim of health insurance is to protect households from catastrophe or impoverishment [42]. Unfortunately, our findings showed that, while existing health insurance schemes provide some financial protection for migrant worker households, this protection is insufficient. The number of migrant workers’ families with at least one member without basic medical insurance accounted for 20.3%. Therefore, households not covered by basic health insurance would allocate a higher percentage of their capacity to pay for health expenditure [4,43,44].

Similar to previous studies [10,11,12,13,14,32,33], our results also showed that households with members with chronic diseases or those with one or more family members admitted to the hospital were more likely to suffer from CHE. The reason for this is that, not only do chronic diseases or hospitalization limit the economic output of households by limiting a person’s ability to work productively, but they also account for high healthcare expenditure [38,45]. Thus, they may be an important driver in the situation becoming “catastrophic” or resulting in impoverishment [46]. This finding has important implications for policymakers who aim to develop financial and social protection interventions to better protect at-risk migrant worker groups. For example, adding insurance coverage for chronic disease outpatient care or higher reimbursement levels for inpatient services may help to shield patients with chronic diseases or hospitalization from high OOP expenditure or CHE [13].

The occurrence of CHE is the result of many factors. It is difficult to achieve the desired effect by only controlling for a certain aspect. Therefore, studying the relationship between various risk factors is crucial to reduce the occurrence of CHE. MCA, as a statistical description method, cannot test the relationship between variables. Therefore, the combination of MCA and logistic regression can be used to study the influencing factors of CHE, which can be statistically confirmed and visually displayed. The relationship plays a complementary role [28,29,30,31]. This study found that there was a close relationship between CHE and income ≤10,000 Yuan, primary school education or below, and having no basic medical insurance, suggesting that this group of people had a greater risk of CHE, and there was also a strong correlation between these factors. Interactions affect the occurrence of CHE. Affected by urbanization, many rural residents choose to leave the village to work in cities, and they become part of a large special group of migrant workers in China. This part of the rural population mostly has a low education level (85% high school education and below), resulting in lower incomes and a lack of awareness of the risk of not participating in basic health insurance (20% of households without basic health insurance), thereby increasing the risk of CHE.

There were certain strengths and limitations of our study. This was the first study to examine the status and influencing factors of CHE on migrant workers in China or around the world. However, our findings should be interpreted with caution as there may have been a possible recall bias for most of the questionnaire data, especially for the self-reported information on out-of-pocket expenditure, annual household expenditure, and food expenditure, given that asking for a total number may introduce more measurement error compared to using an itemized list. Furthermore, the migrant worker household data only came from cities with a large floating population in Western China, and thus, it may not accurately represent the whole country. The next step is to expand the sample to include cities in both Eastern and Western China.

## 5. Conclusions

The CHE incidence among migrant workers in Western China is severe, and factors that have a strong association with CHE include household per capita annual income, head-of-household level of education, the presence of elderly people over 65 years old or children under five years old, having at least one person in the household with no basic medical insurance, or having chronically ill or hospitalized patients. MCA and logistic regression analysis can be used to evaluate the internal relationships among factors related to CHE. Policymakers should pay more attention to migrant worker households possessing the above characteristics compared to other households so as to decrease the incidence of CHE in this group.

## Figures and Tables

**Figure 1 ijerph-16-00738-f001:**
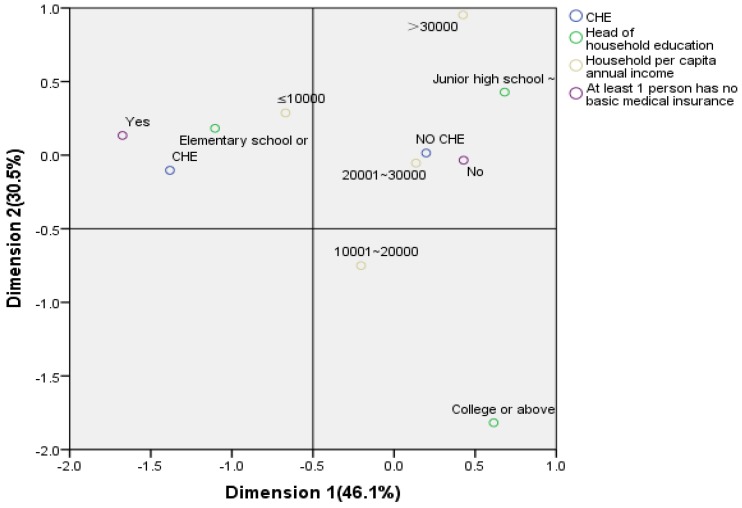
Correspondence plot of the data from migrant workers’ families.

**Table 1 ijerph-16-00738-t001:** Univariable analysis of different migrant workers’ family characteristics (N, %).

Variables	Total (*N* = 1271)	CHE Households (*N* = 159)	χ^2^	*p* Value
**Family sizes (person)**			9.44	0.009
1–2	616 (48.5)	60 (9.7)		
3–4	532 (41.9)	77 (14.5)		
≥5	123 (9.7)	22 (17.9)		
**Household per capita annual income (Yuan)**			24.96	0.000
≤10,000	141 (11.1)	31 (22)		
10,001–20,000	464 (36.5)	70 (15.1)		
20,001–30,000	325 (25.6)	34 (10.5)		
>30,000	341 (26.8)	24 (7)		
**Head-of-household gender**			0.11	0.718
Male	735 (57.8)	90 (12.2)		
Female	536 (42.2)	69 (12.9)		
**Head-of-household age**			3.692	0.297
≤29	224 (17.6)	20 (8.9)		
30–39	316 (24.9)	40 (12.6)		
40–49	401 (31.5)	53 (13.1)		
≥50	330 (26.0)	47 (14.2)		
**Head-of-household education**			103.78	0.000
Elementary school and below	478 (37.6)	67 (14)		
Junior high school–high school	603 (47.4)	79 (13.1)		
University and above	190 (14.9)	15 (7.9)		
**Elderly people over 65 years old**			15.18	0.000
Yes	252 (19.8)	50 (19.9)		
No	1019 (80.2)	108 (10.6)		
**Children under 5 years old**			13.09	0.000
Yes	78 (6.1)	21 (27.1)		
No	1193 (93.9)	138 (11.6)		
**At least 1 person has no basic medical insurance**			38.82	0.000
Yes	258 (20.3)	62 (24)		
No	1013 (79.7)	97 (9.5)		
**Chronically ill patients**			9.01	0.003
Yes	193 (15.2)	37 (19.2)		
No	1078 (84.8)	122 (11.3)		
**Hospitalized patients**			35.75	0.000
Yes	111 (8.7)	34 (30.7)		
No	1160 (91.3)	125 (10.8)		

**Table 2 ijerph-16-00738-t002:** Multivariate logistic regression analysis of CHE of migrant workers’ families.

Variables		Reference Group	Wals χ^2^	*p* Value	OR	95% CI
**Family sizes (person)**	3–4	1–2	0.19	0.662	0.88	0.49–1.58
	≥5		0.05	0.826	1.06	0.61–1.84
**Household per capita annual income (Yuan)**	20,001–30,000	>30,000	2.42	0.120	1.54	0.89–2.67
	10,001–20,000		11.80	0.001	2.35	1.44–3.82
	≤10,000		20.05	0.000	3.72	2.09–6.62
**Head-of-household education**	Junior high school–high school	University and above	0.00	0.947	0.98	0.47–2.03
	Elementary school and below		26.97	0.000	5.90	3.02–11.5
**Elderly people over 65 years old**	Yes	No	14.72	0.000	2.05	1.42–2.97
**Children under 5 years old**	Yes	No	12.26	0.000	2.61	1.53–4.48
**At least 1 person has no basic medical insurance**	No	Yes	36.31	0.000	2.96	2.08–4.23
**Chronically ill patients**	Yes	No	8.79	0.003	1.85	1.23–2.77
**Hospitalized patients**	Yes	No	32.13	0.000	3.61	2.31–5.62

**Table 3 ijerph-16-00738-t003:** Correlation analysis of the research variables.

CHE	Household Per Capita Annual Income	Head-of-Household Education	Basic Medical Insurance
Head-of-household education	11.22 **	-	39.65 **
Basic medical insurance	45.62 **	39.65 **	-
CHE	24.96 **	103.78 **	38.82 **

The results display the chi-square values. ** *p* < 0.05.

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
