# Peer review of "Empirical Analysis of the Status and Influencing Factors of Catastrophic Health Expenditure of Migrant Workers in Western China"

_ijerph, 2019, doi:10.3390/ijerph16050738_

Round 1

Reviewer 1 Report

This article documents the incidence of catastrophic health expenditure among migrant workers in western China and provides analyses of the factors that might affect the CHE incidence. This work is valuable to the literature. However, the writing of the article hinders its contribution. The discussion can also be improved to explain the significance of the findings.

1.     In the abstract, the Results section is poorly written and contains several grammar mistakes. Please revise. The writing of the Discussion section is also unclear.

2.     The writing of the Introduction section is consistent and clear. It provides good motivation for why studying CHE among migrant workers is important. I would suggest revising the writing of the whole paper similar to the standard of the Introduction.

3.     In the materials and methods section, because this paper uses a non-public data source, I would like to see more description of the data collection. For example, how the “residential houses” can be identified as migrants v.s. local urban residents. What is a “cluster” in the sampling? If some evidence as for how the average age of the sample is similar to the national figure, it could increase the confidence in the representativeness of this sample.

4.     There are also several questionable writings in the materials and methods section, such as “all households of migrant workers” (does this mean all members in one household are interviewed or what?)

5.     The details of how family income and expenditures are asked are very important to assess data quality. Just asking a total number introduces much more substantial measurement errors compared to an itemized list. Providing this piece of information is very helpful for readers to understand the quality of the CHE measure and other key factors.  

6.     The Results section is poorly written and requires substantial revision. It also has many typos. For example, Chi-squared is not superscripted. The tables are more clear compared to the writing.

7.     For the multivariate analysis, the paper did not explain what other control variables are included, like the age and gender of the household head. Are they included but not significant, or not included at all? This has implications on the education level explanation, as having lower than primary education is closely correlated with older household heads.

8.     The Discussion section can provide more depth to the findings. For example, the comparison of the CHE among migrant and urban households can be more substantial: what do the level difference mean and how does it compare to WHO standard etc. 

Author Response

 Point 1:In the abstract, the Results section is poorly written and contains several grammar mistakes. Please revise. The writing of the Discussion section is also unclear.

Response 1: The article has undergone English language editing by MDPI. The Discussion section has been revised.

Point 2: The writing of the Introduction section is consistent and clear. It provides good motivation for why studying CHE among migrant workers is important. I would suggest revising the writing of the whole paper similar to the standard of the Introduction.

Response 2: The whole paper has undergone English language editing by MDPI.

Point 3: In the materials and methods section, because this paper uses a non-public data source, I would like to see more description of the data collection. For example, how the “residential houses” can be identified as migrants v.s. local urban residents. What is a “cluster” in the sampling? If some evidence as for how the average age of the sample is similar to the national figure, it could increase the confidence in the representativeness of this sample.

 Response 3:

1.In “Materials and Methods”, after revision, the multi-stage stratified cluster random-sampling technique has been described in detail.

2.“Residential houses” were identified as housing migrants versus local urban residents, mainly by asking the following questions: (1) Whether the household registration was in the rural areas; (2) whether they had entered the city to work in non-agricultural labor; and (3) whether they had been working in non-agricultural industries for more than 6 months. If the above questions were answered with “yes”, the household was included in our research and vice versa.

3.The buildings of residential areas is a “cluster” in the sampling.

        4.The national figure about  the age of the head of the migrant worker’s family has not been found for the time being.

Point 4:There are also several questionable writings in the materials and methods section, such as “all households of migrant workers” (does this mean all members in one household are interviewed or what?)

Response 4: The questionnaire was answered by the member most familiar with the family’s situation. The article has modified this issue.

Point 5:The details of how family income and expenditures are asked are very important to assess data quality. Just asking a total number introduces much more substantial measurement errors compared to an itemized list. Providing this piece of information is very helpful for readers to understand the quality of the CHE measure and other key factors.

Response 5: We asking a total number. At the same time, it was mentioned in the limitations of the article.

Point 6: The section is poorly written and requires substantial revision. It also has many typos. For example, Chi-squared is not superscripted. The tables are more clear compared to the writing.

Response 6: The Results section has been rewritten.

Point 7: For the multivariate analysis, the paper did not explain what other control variables are included, like the age and gender of the household head. Are they included but not significant, or not included at all? This has implications on the education level explanation, as having lower than primary education is closely correlated with older household heads.

Response 7: They are included but not significant. After revision, in the univariable analysis, the two variables have been stated and displayed, they are not significant.

Point 8:The Discussion section can provide more depth to the findings. For example, the comparison of the CHE among migrant and urban households can be more substantial: what do the level difference mean and how does it compare to WHO standard etc. 

Response 8 : The Discussion section has been revised and provided more depth to the findings.

Reviewer 2 Report

The manuscript describes a very interesting and actual topic and fots the Journal's scope. However, several modifications should be done before its acceptance, as detailed below:

- The Abstract needs to be rewritten, especially in its central part. In it, Results are described in a confused way, with several typos and errors. The local currency should be specified at the first mention, not after it.

- The Introduction is overall clear, with an adequate reference to the relevant literature. Just the initial and the final parts are a little bit confusing. Please, re-write those parts.

- In the Methods, the ethical approval statement is missing. Please, provide.

- In the Results, the first sentence of 3.2 is not clear. Please, rephrase. Overall, much of the presentation of results is not clear. It is good that tables and figures have been added, but the readability of this section should be significantly improved.

- In the Discussion, many sentences are difficult to read. Please, revise it carefully. Furthermore, some of the conclusions stated appear to be quite speculative, therefore needing to be more related to literature findings. Future developments are missing and should be added.

- Many references are wrongly cited. Please, modify according to the journal's guidelines.

- Overall, English grammar and language is quite poor. Many typos are present. Please, revise.

Author Response

Point 1:The Abstract needs to be, especially in its central part. In it, Results are described in a confused way, with several typos and errors. The local currency should be specified at the first mention, not after it.

Response 1 : The Abstract has been rewritten.

Point 2:The Introduction is overall clear, with an adequate reference to the relevant literature. Just the initial and the final parts are a little bit confusing. Please, re-write those parts.

Response 2 : Those parts has been rewritten.

Point 3:In the Methods, the ethical approval statement is missing. Please, provide.

Response 3 : In the Methods, Ethics Statement has been added.

Point 4:In the Results, the first sentence of 3.2 is not clear. Please, rephrase. Overall, much of the presentation of results is not clear. It is good that tables and figures have been added, but the readability of this section should be significantly improved.

Response 4: The Results section has been rewritten including the first sentence of 3.2.

Point 5:In the Discussion, many sentences are difficult to read. Please, revise it carefully. Furthermore, some of the conclusions stated appear to be quite speculative, therefore needing to be more related to literature findings. Future developments are missing and should be added.

Response 5: The Discussion section has been revised and provided more depth to the findings. we has been added strengths and limitations section in our study, The next step development has made a statement in it.

Point 6:Many references are wrongly cited. Please, modify according to the journal's guidelines.

Response 6: The references section has been modified according to the journal's guidelines.

Point 7: Overall, English grammar and language is quite poor. Many typos are present. Please, revise

Response 7: The article has undergone English language editing by MDPI.

Reviewer 3 Report

I have reviewed this manuscript, and would like to comment on the following weaknesses that will have to be addressed before it could be realistically considered for publication.

·  More background on the context or already available research concerning the presence of high or catastrophic OOP spending in developed OECD countries both pre and post the financial crisis (the homogenous country group).

·  The presented explanations and definitions of catastrophic health expenditures or high OOP spending in the introduction section are poor.

·  In addition, please make the research questions and objectives of your study prominently (distinct) displayed and not as a last part of the introductory section. You should firstly provide your main aim, any research questions-hypotheses and finally the contribution of your study to the existing literature in comparison with previous published studies, concerning your topic for PHI impact on OOP payments levels. In the section Measures-CHE you do not cite any reference!!!!

·  Major changes to the discussion (and conclusions – are necessary to be included) to make them relevant for policy-makers and an international audience.

Author Response

Point 1:More background on the context or already available research concerning the presence of high or catastrophic OOP spending in developed OECD countries both pre and post the financial crisis (the homogenous country group).

Response 1: The first paragraph of the introduction section has been has been rewritten.

Point 2:The presented explanations and definitions of catastrophic health expenditures or high OOP spending in the introduction section are poor.

Response 2: The explanations and definitions of catastrophic health expenditures has been revised.

Point 3:In addition, please make the research questions and objectives of your study prominently (distinct) displayed and not as a last part of the introductory section. You should firstly provide your main aim, any research questions-hypotheses and finally the contribution of your study to the existing literature in comparison with previous published studies, concerning your topic for PHI impact on OOP payments levels. In the section Measures-CHE you do not cite any reference!!!!

Response 3: The introduction section has been rewritten. The method of Measures-CHE has been re-introduced and cited references.

Point 4: Major changes to the discussion (and conclusions – are necessary to be included) to make them relevant for policy-makers and an international audience.

Response 4: The Discussion section has been revised and conclusions section has been added.

Round 2

Reviewer 1 Report

The language has been significantly improved. The discussion section also offers more depth to the results. 

Reviewer 2 Report

The paper has been significantly improved with respect to the previous version. The help of the editing tools is evident, making the paper much more clear now. However, despite this, several typos are still present and need to be carefully revised.

Reviewer 3 Report

Accept in present form